# The Expression and Secretion Profile of TRAP5 Isoforms in Gaucher Disease

**DOI:** 10.3390/cells13080716

**Published:** 2024-04-20

**Authors:** Margarita M. Ivanova, Julia Dao, Neala Loynab, Sohailla Noor, Neil Kasaci, Andrew Friedman, Ozlem Goker-Alpan

**Affiliations:** Lysosomal and Rare Disorders Research and Treatment Center, Fairfax, VA 22030, USA; jdao@ldrtc.org (J.D.); nkasaci@ldrtc.org (N.K.); ogoker-alpan@ldrtc.org (O.G.-A.)

**Keywords:** Gaucher disease, osteoporosis, TRAP, macrophages, Gaucher cells, bone, inflammation

## Abstract

Background: Gaucher disease (GD) is caused by glucocerebrosidase (GCase) enzyme deficiency, leading to glycosylceramide (Gb-1) and glucosylsphingosine (Lyso-Gb-1) accumulation. The pathological hallmark for GD is an accumulation of large macrophages called Gaucher cells (GCs) in the liver, spleen, and bone marrow, which are associated with chronic organ enlargement, bone manifestations, and inflammation. Tartrate-resistant acid phosphatase type 5 (TRAP5 protein, *ACP5* gene) has long been a nonspecific biomarker of macrophage/GCs activation; however, the discovery of two isoforms of TRAP5 has expanded its significance. The discovery of TRAP5′s two isoforms revealed that it is more than just a biomarker of macrophage activity. While TRAP5a is highly expressed in macrophages, TRAP5b is secreted by osteoclasts. Recently, we have shown that the elevation of TRAP5b in plasma is associated with osteoporosis in GD. However, the role of TRAP isoforms in GD and how the accumulation of Gb-1 and Lyso-Gb-1 affects TRAP expression is unknown. Methods: 39 patients with GD were categorized into cohorts based on bone mineral density (BMD). TRAP5a and TRAP5b plasma levels were quantified by ELISA. ACP5 mRNA was estimated using RT-PCR. Results: An increase in TRAP5b was associated with reduced BMD and correlated with Lyso-Gb-1 and immune activator chemokine ligand 18 (CCL18). In contrast, the elevation of TRAP5a correlated with chitotriosidase activity in GD. Lyso-Gb-1 and plasma seemed to influence the expression of ACP5 in macrophages. Conclusions: As an early indicator of BMD alteration, measurement of circulating TRAP5b is a valuable tool for assessing osteopenia–osteoporosis in GD, while TRAP5a serves as a biomarker of macrophage activation in GD. Understanding the distinct expression pattern of TRAP5 isoforms offers valuable insight into both bone disease and the broader implications for immune system activation in GD.

## 1. Introduction

Gaucher disease (GD) (Online Mendelian Inheritance in Man (OMIM) 23080, 231000, 231005) is a genetic lysosomal storage disorder caused by the pathogenic variants of the glucocerebrosidase (*GBA1*) gene and, as a result, a deficiency of the glucocerebrosidase enzyme (GCase). GD is clinically divided into non-neuronopathic and neuronopathic forms; the presentations can range from severe progressive (GD2) to subacute neurological disease (GD3) in the pediatric age group [1]. The non-neurological form of GD exhibits a wide range of symptoms, including an enlarged spleen and liver, anemia, low thrombocyte counts, and affecting the skeleton development and bone structure. Bone pain or bone crises, avascular necrosis, bone marrow infiltration, and the development of osteoporosis at an early age frequently occur in all clinical forms of GD [2,3,4].

Due to a GCase enzyme deficiency, the last glycolipid in the catabolic pathway of the glycosphingolipid metabolism, glucosylceramide (Gb-1), is not completely degraded in GD. As a result, excessive levels of Gb-1 and its metabolite glucosylsphingosine (Lyso-Gb-1) accumulate in lysosomes and interfere with cellular pathways, including autophagy-lysosomal and mitochondrial functions [5,6,7,8,9]. The accumulation of Gb-1/Lyso-Gb-1 in macrophages leads to the development of large and foamy macrophages, sometimes forming multinucleated cells [10,11,12]. These macrophages, known as Gaucher cells (GCs), can be detected in the liver, spleen, and bone marrow and are actively involved in chronic immune activation, organomegaly, and skeletal involvement [13]. Chitotriosidase (Chito) and CCL18 are secreted from the activated macrophages and utilized as biomarkers in GD to monitor therapy and assess disease activity [14].

In 1991, spleen biopsies from GD patients showed strong TRAP activity through immunohistochemistry, which later became a nonspecific marker of sphingolipid storage. However, over time, it was discovered that TRAP5 is a metalloprotein enzyme with two secreted isoforms, 5a and 5b, with distinct functions. TRAP5a is a monomer pro-enzyme form without phosphatase activity. TRAP5b is a heterodimer isoform consisting of the posttranslational cleavage of N-terminal fragment 20–23 kDa and a 17-kDa C-terminal fragment with significantly increased phosphatase activity [15,16]. Moreover, TRAP5a and TRAP5b are secreted from different cells. TRAP5a is predominantly expressed in macrophages and dendritic cells and is considered to be a biomarker for inflammatory macrophages [17]. TRAP5b is expressed in osteoclasts and has a significant role in regulating bone resorption and osteoclast migration; as a result, TRAP5b has become a prominent diagnostic biomarker for bone pathology [17,18,19].

In the past, the enzyme-linked immunosorbent assay (ELISA) used an antibody that recognized both isoforms of TRAP, which resulted in the nonuniform detection of TRAP5 in GD, which was not easy to interpret and showed analytical variability [14,19,20]. However, today, the diagnostic purpose of TRAP5 is reinforced with new commercially available TRAP isoform-specific ELISAs that can provide a diagnostic tool with greater accuracy [21].

The primary aim of this study was to develop blood-based biomarkers that could predict the abnormalities in BMD in patients with GD earlier than traditional radiographic methods. Focusing on circulating TRAP5a and TRAP5b as a potential indicator of immune activation and bone resorption, we examined their expression and secretion in peripheral blood and macrophages, correlating these finding with established GD-specific biomarkers (Lyso-Gb-1, chito, and CCL18). Additionally, by comparing the bone resorption marker TRAP5b and the macrophage activation marker TRAP5a against the BMD, we assessed the influence of age on the development of osteoporosis in GD.

## 2. Materials and Methods

### 2.1. Subjects

Under Institutional Review Board (IRB)-approved protocols NCT04055831 and NCT02000310, 39 participants (29 women and 10 men) were recruited. Healthy controls were enrolled under the NCT02000310 protocol, or plasma was purchased (StemExpress, Folsom, CA, USA). The diagnosis of GD was determined based on GCase residual activity and the sequencing analysis of *GBA1*. In the GD cohort, 3 GD patients were treatment-naïve, 22 patients were on enzyme replacement therapy (ERT), 13 were on substrate reduction therapy (SRT), and 1 switched to ERT from SRT. The *GBA1* genotypes included 17 patients with the homozygous p.N370S (N409S). The second most common allele was p.L444P (L483P) (n = 12); one patient had the L444P/L444P genotype (Appendix A [19,22]). In the GD cohorts, the age range was from 18 to 68 years, with an average age of 42 ± 15 years. The average age range in healthy controls was 48 ± 11 years. The patients with GD were categorized into three groups based on their BMD [19]. The patients with GD were categorized into the following cohorts based on BMD: patients with normal BMD (NB; n = 11), patients with osteopenia (OSN; n = 14), and patients with osteoporosis (OSR; n = 14).

### 2.2. Blood Sample Collection

Venous blood samples were collected into EDTA K+ tubes. The whole blood was then centrifuged at 800× *g* for 5 min to separate the plasma. The plasma was then aliquoted into 1.5 mL tubes and spun down at 2000× *g* for 2 min to clear samples. After centrifugation, the plasma was collected and aliquoted into smaller volumes. The remaining blood was diluted with PBS/2% FBS to purify peripheral blood mononuclear cells (PBMCs). Fresh PBMCs were isolated using SepMate^TM^ PBMCs isolation tubes (Stemcells Technology, Cambridge, MA, USA), following the manufacturer’s protocol. Plasma samples were stored at −80 or −120 °C.

### 2.3. Differentiation of M2 Macrophages from PBMCs

For macrophage differentiation, freshly isolated PBMCs were resuspended in an appropriate amount of complete M2 macrophage generation media (RPMI 1640 supplemented with 10% FBS, 1% Normocin, 2 mM glutamine, 1% Na-pyruvate, 1% Non-Essential Amino Acids (NEEA), and 50 ng/mL human recombinant M-CSF (ThermoFisher Scientific, Rockford, IL, USA). After six days of culture, M2 macrophage differentiation media was added. After an additional two days, culture media was replaced entirely with new M2 macrophage differentiation media [23]. On day ten, macrophages were collected for analysis or treated with plasma for 48 h. All cell culture experiments were maintained at 37 °C and 5% CO_2_.

### 2.4. Differentiation of Macrophages from THP-1

The THP-1 cell line (human monocytic cell line derived from an acute monocytic leukemia patient) was obtained from MilliporeSigma (MilliporeSigma, Burlington, MA, USA). Cells were maintained in RPMI-1640 medium supplemented with 10% FBS and 2 mmol/L L-glutamine. THP-1 cells were differentiated to macrophages using 100 nM phorbol 12-myristate 13-acetate (PMA, MilliporeSigma, Burlington, MA, USA) for 72 h. After the PMA stimulation, the PMA-containing media was removed, and the THP-1-macrophages were incubated in fresh RPMI 1640 (10% FBS, 1% L-glutamine) for an additional 24 h.

### 2.5. Cell Treatment

Macrophages derived from healthy controls, GD patients, and THP-1 macrophages were treated with different concentrations of Lyso-Gb-1 (HY-N7745, MedChemExpress, Monmouth Junction, NJ, USA), as indicated in the figures. After treatments, cells were washed with PBS, and RNA lysis buffer was added; cells were collected and stored at −80 °C until analysis.

### 2.6. RNA Isolation and Quantitative Real-Time PCR (q-RT-PCR)

RNA was extracted using the Quick-RNA Microprep Kit (Zymo Research, Irvine, CA, USA). The LunaScript^®^ Multiplex One-Step or Two-step RT-PCR Kit (New England Biolabs, Ipswich, MA, USA) was used for real-time PCR. Compared with reference gene *GAPDH*, mRNA transcript levels were measured using the StepOnePlus™ Real-time PCR System (ThermoFisher Scientific, Rockford, IL, USA). The *ACP5, CCL2/MCP-1*, and *GAPDH* primers were purchased from Eurofins Scientific (Luxembourg, FR, Appendix A). Chitotriosidase (protein abbreviation Chito, gene: Chitinase 1 with abbreviation *Chit1*) primers were purchased (OriGene, Rockville, MD, USA). Gene expression was determined in triplicate and normalized using *GAPDH*. Analyses and fold differences were determined using the comparative CT method. Fold change was calculated from the ΔΔCT values with the formula 2−ΔΔCT, and data are relative to untreated controls.

### 2.7. Measurement of TRAP5 Isoform and Macrophage Activation Biomarkers in Plasma

Bone markers in plasma were measured using commercially available ELISA kits. The concentration of TRAP5b was measured using a MicroVue^TM^ TRAP5b EIA kit (Quidel Corporation, San Diego, CA, USA). The calibration was carried out using recombinant TRAP5b, and the assay range was 0–16.5 U/L. The concentration of TRAP5a was measured using TRAP5a kits (MyBioSource, San Diego, CA, USA). Recombinant TRAP5a was used as a standard; the assay range was 2.5–50 ng/mL and the analytical sensitivity was 1.0 ng/mL. The concentration of CCL18 was measured using a PARC/CCL18 Human ELISA Kit (ThermoFisher Scientific, Waltham, MA, USA), according to the manufacturers’ protocols.

### 2.8. Analysis of Chitotriosidase Enzymatic Activity in Dried Blood Spot (DBS) Samples

A chitotriosidase activity assay (Chito) was performed using DBS, as previously described [20,24]. The concentrations of the substrate 4-methylumbelliferyl-β-D-N,N′,N″-triacetyl-chitotrioside (4MU-C3, Sigma^®^ MilliporeSigma, Burlington, MA, USA) were 0.11 mM in 0.1 M/0.2 M citrate-phosphate buffer. The reaction was incubated for 3 h and stopped with Glycine-Sodium Hydroxide Buffer. The signal was measured with an emission of 360 nm and excitation of 450 nm using Spectramax M2 (Molecular Devices, LLC, San Jose, CA, USA). The unit of the enzyme activity is expressed as nmol/hr/mL.

### 2.9. Statistical Analysis

Statistical analysis was conducted using GraphPad Prism (GraphPad, San Diego, CA, USA). Student’s *t*-tests or F-tests were utilized to analyze differences between the two groups. The groups were compared using a one-way analysis of variance (ANOVA), followed by Kruskal–Wallis tests. A *p*-value of less < 0.05 indicated a statistically significant result. Correlation analysis between the two groups was performed using either the Pearson or Spearman correlation technique.

## 3. Results

### 3.1. The Assessment of TRAP5a and TRAP5b Plasma Levels in GD Cohorts

We previously reported that plasma TRAP5b levels are increased in patients with GD, associated with osteopenia and osteoporosis, and positively correlated with GD clinical biomarkers CCL18, Lyso-Gb-1, and Chito [19]. Due to the activation of macrophages in GD, we also anticipated elevated levels of TRAP5a in patients’ plasma [11,13,25]. However, TRAP5a was elevated in only 15 of 40 patients with GD. The mean level of TRAP5a in GD patients was 13.1 ± 1.9, compared to 8.9 ± 0.8 in healthy controls. The difference in means was 4.5 ± 2.3 (Figure 1A). Further analysis showed no correlation between TRAP5a and osteopenia/osteoporosis, unlike TRAP5b (Figure 1B).

Next, we validated the relationship between TRAP5a plasma level and BMD in GD patients with different therapies, including enzyme replacement therapy (ERT) and substrate reduction therapy (SRT). However, the group of patients who were not receiving any treatment (NAÏVE GD patients) was excluded from statistical analysis due to their small number (Figure 1C). There was a significant increase in the level of TRAP5a in the cohort of “Normal BMD” who received SRT only compared to the control group (Figure 1C). However, the study’s limited number of patients may constrain the interpretability of the data, particularly in understanding the variability in TRAP5a levels. For instance, a naïve patient with OSN and an elevated level of TRAP5a (62.8 ng/mL) also exhibited increased Chito (1194 nmol/hr/mL), Lyso-Gb-1 (49 ng/mL), and CCL18 (407 ng/mL). Conversely, two other untreated GD patients with OSR did not show dramatic increases in these biomarkers, Chito (38 and 20 nmol/hr/mL), Lyso-Gb-1 (2.5 and 5.4 ng/mL), and CCL18 (45 and 13 ng/mL), indicating that patient-specific factors may influence these levels and their relationship to GD manifestations.

With the increasing number of GD patients in our study, the TRAP5b levels were measured in new patients, and all samples were reanalyzed. Similar to our previous data [19], elevated levels of TRAP5b in GD correlate with osteopenia–osteoporosis (Figure 1D,E). Assessment of the relationship between plasma TRAP5a and BMD in patients on different therapies demonstrated that neither ERT nor SRT impacted the elevation of TRAP5b (Figure 1E). Furthermore, analysis of the relationship between TRAP5a and TRAP5b revealed the absence of a linear correlation between TRAP isoforms in GD (Figure 1G). This finding could suggest that the regulation of TRAP isoforms is controlled by separate factors and underscores the unique roles and regulatory mechanisms of TRAP5 isoforms contributing to GD pathology.

### 3.2. Long-Term Monitoring of TRAP5a and TRAP5b in Patients with GD

Most GD patients with normal BMD or osteopenia did not show any significant changes in TRAP5a levels over 24 months, except for two patients with normal BMD and two patients with osteopenia (Figure 2A,B). One patient with normal BMD exhibited an increased TRAP5a level, whereas in other patients the TRAP5a level decreased over the same period. Additionally, the TRAP5b level remained relatively stable over two years and did not show any significant changes in patients with normal BMD (Figure 2D,E). Only two patients showed a slight elevation in TRAP5b levels. Among GD patients with osteopenia, the TRAP5b level remained unchanged in five patients, increased in one patient, and significantly decreased in another patient (Figure 2F). Most GD patients with osteoporosis had elevated TRAP5b levels, except for two patients who showed normal levels on visit one but elevated levels 24 months later (visit 5). Additionally, one patient showed a normalization of the TRAP5b level, and the BMD test showed a stabilization of the T-score over 24 months (Figure 2F).

### 3.3. Correlation Analysis between TRAP5a, TRAP5b, and Other Inflammatory Biomarkers in GD

Then, we assessed the correlation between TRAP5a, TRAP5b, and Chito and CCl18 inflammatory biomarkers elevated in GD. Similar to TRAP5, Chito (gene abbreviation CHIT-1) is highly expressed in activated macrophages and Kupffer cells, while CCL18 is produced by dendritic cells, monocytes, and macrophages [26,27]. While TRAP5b positively correlates with Chito and CCL18 in the GD cohort [19], Spearman rank correlation analysis of TRAP5a and Chito in all GD patients showed heteroscedasticity where the level of TRAP5a or Chito increased (Pearson correlation *p* = 0.053 and Spearman correlation *p* = 0.016, Figure 3A,D–F). This heteroscedastic distribution can be explained by the different regulatory mechanisms of TRAP5a and Chito expression in macrophages under chronic inflammation [26,28,29]. CCL18 was found to be associated with TRAP5b, but it did not show any correlation with TRAP5a (Figure 3B,D–F).

Next, we compared the levels of TRAP5a and TRAP5b with the levels of Lyso-Gb-1, the most reliable diagnostic and pharmacodynamics biomarker for GD [30,31,32]. There was a strong positive correlation (r = 0.8) between TRAP5b and Lyso-Gb1, while no correlation was found between TRAP5a and Lyso-Gb-1 (r = 0.002) (Figure 3C–F).

### 3.4. The Role of TRAP5b, TRAP5a, and the Bone Mineral Density Abnormalities in GD

The bone aging process involves a decreasing BMD, which leads to predisposing primary osteoporosis [33,34]. In the general population, women ≥ 50 and men ≥ 60 start losing BMD, with a rapid decline occurring within 65–69 years for women and 74–79 years for men [35]. However, for GD patients, a significant decrease in BMD occurs with the early onset of osteoporosis, observed not only in untreated patients but also in patients under ERT or SRT. Therefore, it was not surprising that the patients in our cohort with GD, who had OSN and OSR, were much younger than those with OSR in the general population (Appendix A). Correlation analysis of age and Z- or T-score in GD showed that a decreasing BMD does not correlate with age (Figure 3D–F and Figure 4A,B). It is of note that 41% of GD patients with OSN and OSR included in the study were under 50 years old. A total of 16 out of 28 individuals with OSN or OSR were under 50, including 10 women and 6 men. The median ages for females with OSN and OSR are 46 and 50, respectively, and for males the median ages for OCN and OSR are 29 and 41, respectively (Appendix A). The youngest patient diagnosed with GD and osteoclastic lesions was an 18-year-old male with the N370S/N370S *GBA1* variant. The patient was not undergoing any therapy at the time and exhibited bone marrow infiltration. In summary, decreasing BMD in GD is unrelated to age (Figure 3D–F and Figure 4C,D); thus, predicting the risk of osteoporosis development based on age is not effective for GD patients. Furthermore, the elevation of TRAP5b and TRAP5a is not correlated with patients’ age (Figure 3D–F and Figure 4C,D). However, it has been observed that TRAP5b, and not TRAP5a, shows a positive correlation with the Z-score (Figure 3D–F and Figure 4E,G).

### 3.5. ACP5 mRNA Expression in Cultured GD Macrophages

Next, we examined the expression of TRAP (gene abbreviation ACP5) mRNA in macrophages from healthy control and GD patients. In comparison, MCP-1 and Chit-1 mRNA levels were also evaluated. To investigate the mRNA expression profile, M2 macrophages were differentiated from PBMCs isolated from healthy controls and GD patients (Table 1). M2 macrophages were derived from seven GD patients, five patients with N370S/N370S genotype, one with N370S/Y412X, and one with L444P/L444P (Table 1). The clinical data of patients are presented in Table 1, including osteopenia, osteoporosis status, GD biomarkers (lyso-Gb1, CCL18, and Chito), and inflammatory biomarkers (GM-CSF, TNF-alpha, and CCL2/MCP-1). In addition, the elevated plasma biomarker levels are highlighted in the table (Table 1). The high TNF-alpha in the patient’s cohort agrees with previously published data concerning the elevation of TNF-alpha in GD [36,37].

As shown in Figure 5A, ACP5 levels were significantly decreased in cultured GD macrophages after differentiation for 12 days, except for one macrophage cell line derived from GD patients with osteoporosis (P7). A similar result was detected with CCL2/MCP-1 mRNA (Figure 5B). Since the differentiation and culture of macrophages were performed under the same conditions for controls and GDs, only a deficiency of GCase could be a factor that suppresses the mRNA expression of ACP5 and CCL2/MCP-1 in macrophages. A decreased expression of some cytokines in cultured GD macrophages has been described for CCL5/RANTES and CXCR4, while the serum level of CCL5/RANTES is elevated in GD patients with osteonecrosis [38]. In contrast to ACP5 and CCL2/MCP-1, Chit-1 mRNA was significantly elevated in GD macrophages (Figure 5C).

To examine the contribution of GD plasma on the expression of endogenous ACP5, macrophages differentiated from healthy controls and GD were treated with plasma, including plasma from patients with normal BMD (ND, P2), OSN (P23), and OSR (P35 and P37) (Figure 5B). GD plasma induced ACP5 mRNA expression in control and GD macrophages (Figure 5B,C). In contrast to ACP5, the CCL2/MCP-1 and Chit1 expression did not change in the presence of control or GD plasma (Figure 5D,E). Thus, the expression of ACP5 induced by plasma shows that external factors present in the blood play a role in the regulation of ACP5 expression.

Next, we investigated the role of Lyso-Gb-1 levels in the regulation of ACP5 transcription in macrophages from patients with GD. Treatment of macrophages with Lyso-Gb-1 induced the ACP5 expression level in a concentration- and time-dependent manner (Figure 5F). Additionally, we would like to mention that the activation of ACP5 mRNA expression was very robust after just 1 h of treatment with 0.2 and 2 μM of Lyso-Gb-1 (Figure 5F).

### 3.6. Regulation ACP5 Expression and Lyso-Gb-1

Next, we used macrophages derived from human monocytic leukemia THP-1 cells as an alternative model for macrophages to validate the role of Lyso-Gb-1 in ACP5 expression. THP-1 cells differentiated into macrophages were treated with increasing concentrations of Lyso-Gb-1. The results showed that ACP5 mRNA expression increased significantly in the presence of Lyso-Gb-1, similar to the results obtained from macrophages derived from GD patients (Figure 5G). The data also revealed that Lyso-Gb-1 activates TRAP5 expression, commensurate with previous reports of the activation of ACP5 (TRAP5) in CBE-treated cellular models [39].

## 4. Discussion

Although TRAP has historically been used in GD as a clinical biomarker [10,40], we still do not fully understand the role of TRAP5a and TRAP5b in GD pathology and how the accumulation of Gb-1 and Lyso-Gb-1 affects *ACP5* gene expression. The majority of studies examining TRAP as a GD biomarker have relied on antibody-based assays that have been used to quantify the total TRAP5 level in the samples. Since the discovery of TRAP5 isoforms, it became evident that TRAP is not just a biomarker of macrophage activity. With the development of new antibodies that can distinguish the two isoforms, it has become apparent that TRAP5a is an inflammatory biomarker, while TRAP5b is a bone biomarker secreted from osteoclasts [41,42].

TRAP5a is highly expressed in alveolar macrophages, inflammatory macrophages, and biomaterial-induced multinucleated giant cells (BMGCs), and less highly expressed in activated macrophages and dendritic cells [21,29,43,44]. It has been shown that TRAP5a correlates with rheumatoid factors and the severity of rheumatoid arthritis due to macrophage activation and inflammatory loads [44,45]. Furthermore, TRAP5a is associated with pathological adipose tissue expansion and body mass index (BMI) [46]. In the context of GD, the upregulation of TRAP5a can indicate macrophages’ pro-inflammatory activities and an increasing proportion of glycolipid-enriched Gaucher cells. Our data suggest that *ACP5* mRNA expression in macrophages may be stimulated by Lyso-Gb-1, along with other factors present in GD plasma, such as the elevated RANKL in GD [19,47,48]. These results are consistent with previous reports of the activation of ACP5 (TRAP5) in CBE-treated cellular models [39]. The positive correlation between TRAP5a and Chito, but not between TRAP5a and CCL18, suggests that macrophages might be driving pro-inflammatory activities and, possibly, within the GC activity conjunction. Furthermore, our results demonstrated that an increased level of TRAP5a in GD is not correlated with TRAP5b or decreasing BMD, and thus does not correlate with osteoclast activity and the acceleration of bone resorption.

Although the role of TRAP5b in bone metabolism is well described, the function of TRAP5a in macrophages is still not fully defined. In addition, little is known about the regulation of TRAP expression in macrophages. The TRAP5a isoform is a monomer without enzymatic activity, is highly expressed in alveolar macrophages, and is linked to NO production as a mechanism of protection from bacterial infection [49]. Furthermore, TRAP5a enhances macrophage migration in lung tissue, with high expression and activity in the lung tissues of COPD and asthma patients, and is considered a biomarker, reflecting disease activity in chronic inflammatory diseases, including rheumatoid arthritis [29,50]. Several transcription factors control the TRAP promoter and its transcriptional activity. Studies have shown that microphthalmia transcription factor (MITF), nuclear factor of activated T-cells c1 (NFATc1), and receptor activator of NF-κB ligand (RANKL) directly bind to the proximal TRAP promoter to increase its activity. In bone remodeling, RANKL is an essential cytokine for osteoclast activation by promoting TRAP5b expression. Consequently, RANKL indirectly promotes osteoclasts to break down the tissue in bones and, after bone dissolution, release TRAP5b into the extracellular space [51]. Thus, RANKL and components of the RANK pathway, which serve as a bridge between the immune and skeletal systems, may be utilized as markers to track the progression of bone disease in GD patients [19,52].

The degree and the nature of skeletal involvement in GD are variable, often associated with decreased BMD occurring at a young age, and accompanied by significant bone pain, disability, and reduced quality of life. TRAP5b has been described as a highly sensitive and specific diagnostic marker for osteopenia or osteoporosis and is related to bone-associated disorders, e.g., postmenopausal osteoporosis [53], rheumatoid arthritis [22,23], hyperparathyroidism [24], Paget’s disease [54], or bone metastasis [25]. Recently, we demonstrated that TRAP5b could be used as a predictive biomarker of osteoporosis in GD [19].

Moreover, the elevation of TRAP5b, rather than TRAP5a, correlates with a decline in BMD Z-score in patients with GD. In vitro and in vivo studies support the long-standing dispute that circulated TRAP5b represents osteoclast number, not activity [2,19]; therefore, we agree that the elevation of TRAP5b shows increasing bone resorption due to increasing numbers of osteoclasts. Consequently, TRAP5b could serve as an indicator for bone loss, complementing other GD biomarkers such as Gb-1 and Lyso-Gb-1, and can be used for disease monitoring.

## 5. Conclusions

This study is crucial for elucidating the underlying mechanisms involved in bone pathology in GD. TRAP5 isoforms that contribute to the development and progression of BMD abnormalities in GD could be therapeutic targets for developing adjunct therapies. The positive correlations between TRAP5b and GD-specific biomarkers, Lyso-Gb-1, CCL18, and Chito, with the lower BMD Z-score suggest that the accumulation of glucosylsphingosine may play a crucial role in the development of osteoporosis in patients with GD. However, the relationship between TRAP5a and the macrophage activation biomarker Chito suggests that they share a mutual origin of secretion, likely from activated macrophages or Gaucher cells. Consequently, TRAP5a/Chito may serve as an effective biomarker combination to assess macrophage activation status in GD, particularly in the context of immune-driven inflammation.

## Figures and Tables

**Figure 1 cells-13-00716-f001:**
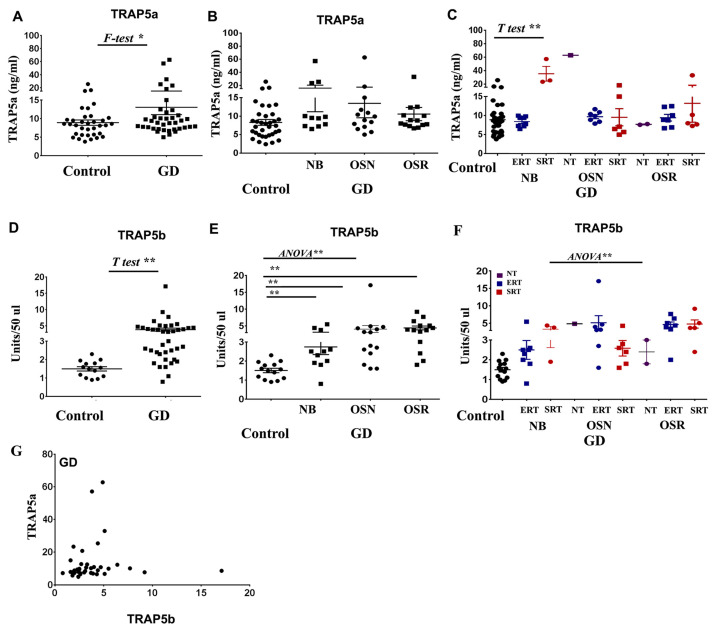
Plasma TRAP5a and TRAP5b concentrations. (**A**) TRAP5a level, Control vs. GD. F-Test, two tail *p* < 0.0001, unpaired *t*-test one tail *p* = 0.0323, two tail *p* = 0.0646, Control n = 35, GD n = 39. (**B**) TRAP5a concentrations in control subjects and GD with no bone complication (NB), osteopenia (OSN), and osteoporosis (OSR). There is no significant difference between NB, OSN, and OSR. Control n = 35, GD-NB n = 11, GD-OSN n = 14, GD-OSR n = 14. (**C**) Relationship between the TRAP5a plasma level and BMD in patients with GD on different therapies: no treatments—NT, NAÏVE, enzyme replacement therapy—ERT, and substrate reduction therapy—SRT. The analysis shows that there is only a significant difference between the control group and patients with normal BMD who are on SRT therapy. The unpaired *t*-test one tail *p* < 0.001. (**D**) TRAP5b level, Control vs. GD. *p* < 0.05 *t*-test. Control n = 14, GD patients: n = 39. (**E**) TRAP5b level in control subjects and patients with GD with no bone complication (N), osteopenia (OSN), and osteoporosis (OSR). ** *p* < 0.05; Kruskal–Wallis’s test, one-way ANOVA. (**F**) Relationship between the level of TRAP5b and BMD in patients with GD who are on different therapies. The Kruskal–Wallis’s test indicates significant differences between the groups (*p* < 0.0001). Dunn’s multiple comparison test shows significant differences between the control group and OSN/ERT, controls and OSR/ERT, and controls and OSR/SRT cohorts. Two-group comparisons using the unpaired *t*-test show significant differences between Control and NB cohorts (SRT and ERT), Control and OSN (SRT and ERT), and Control and OSR (SRT and ERT). Due to the limited number of samples, the “No treatment” OSN and OSR were excluded from the analysis. (**G**) Scatterplot analysis of TRAP5a and TRAP5b levels in patients with GD. Pearson and Spearman correlation analysis determined the absence of correlation between TRAP5a and TRAP5b. Pearson correlation r = 0.06, *p* = 0.35, Spearman correlation r = 0.27, *p* = 0.08. The asterisk (*) *p* ≤ 0.05; (**) *p* ≤ 0.01.

**Figure 2 cells-13-00716-f002:**
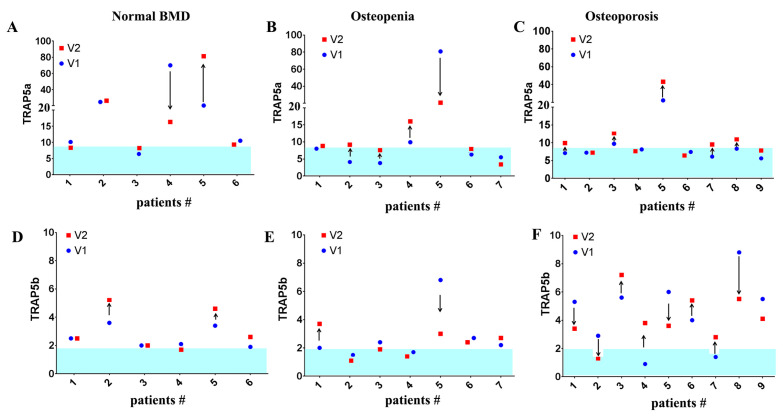
Longitudinal dynamics of TRAP5a and TRAP5b. Visit 1 (V1) is the initial visit, and visit 5 (V5) is the follow-up visit, which is an average of 24 months later. (**A**–**C**) Plasma level of TRAP5a level within 24 months of monitoring. GD patients with normal BMD (**A**), osteopenia (**B**), and osteoporosis (**C**). (**D**–**F**) Plasma level of TRAP5b within 24 months of monitoring. GD patients with normal BMD (**D**), with osteopenia (**E**), and osteoporosis (**F**). Blue color indicates the normal range of TRAP5a or TRAP5b.

**Figure 3 cells-13-00716-f003:**
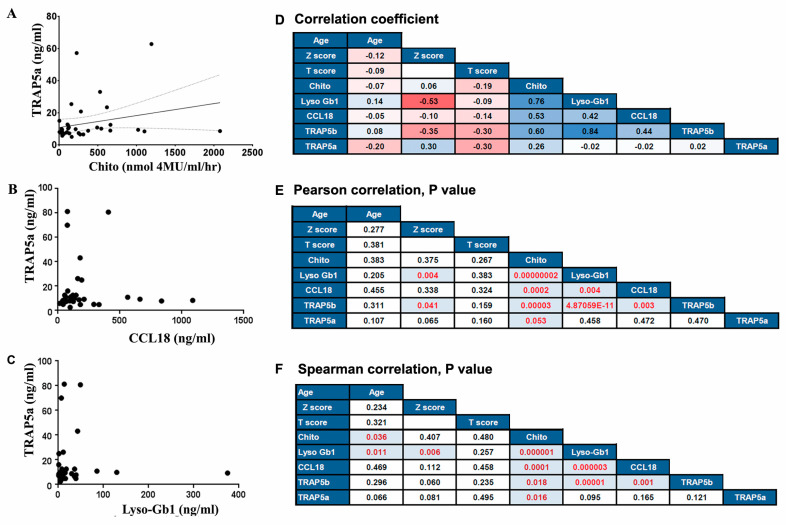
Correlation between TRAP5a and GD clinical biomarkers Chito, Lyso-Gb1, and CCL18. (**A**) Scatterplot analysis of TRAP5a and Chito. The correlation between TRAP5a and Chito was determined by Spearman correlation analysis. *p* < 0.05 was considered statistically significant. (**B**,**C**) Scatterplot analysis of TRAP5a, Lyso-Gb1 (**B**), and CCL18 (**C**) showed no correlation between biomarkers. (**D**–**F**) Correlation matrix and hierarchical clustering. Correlation coefficients for measurements of biomarkers and clinical parameters are visualized by tile-color intensities (blue color, strong; light red color, weak, deep red color, negative correlation). Correlation coefficient = 0.8, strong positive relationships; correlation coefficient = between 0.5 and 0.7, a moderate positive relationship; correlation coefficient between 0.3 and 0.5 indicates variables with a low correlation. Pearson’s correlation (**E**) and Spearman correlation (**F**) *p*-values are labeled inside the titles. The red color indicates significant differences, *p* < 0.05.

**Figure 4 cells-13-00716-f004:**
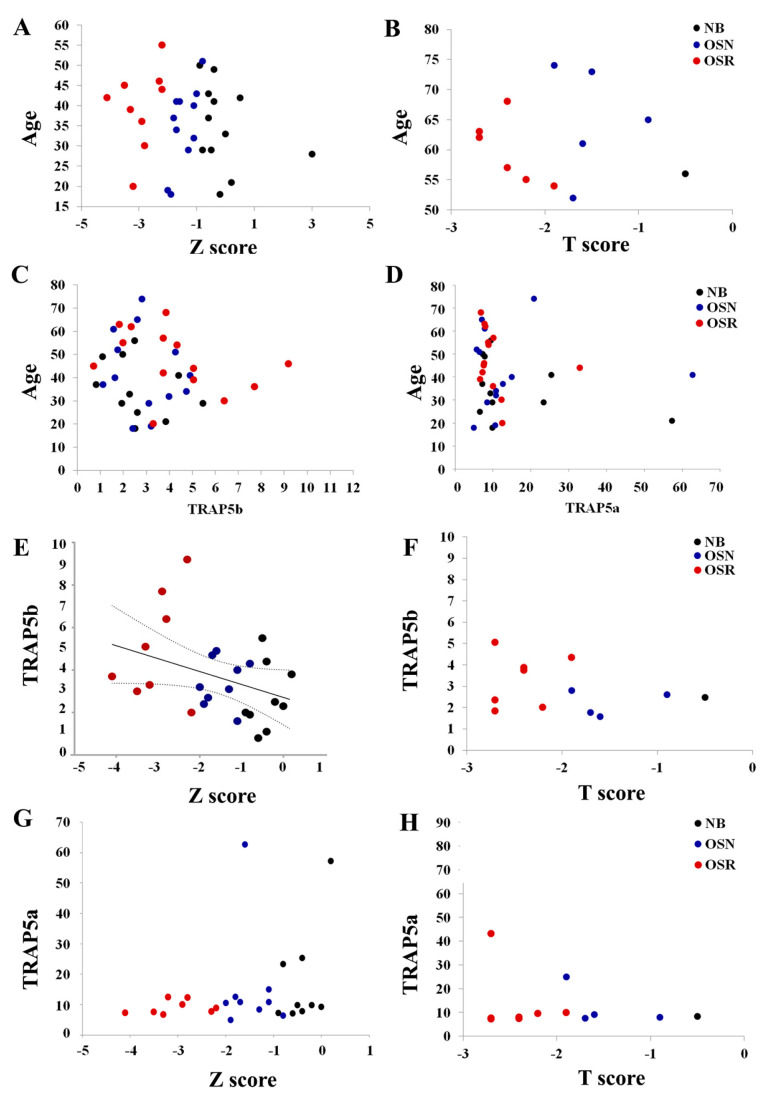
Comparison of age group, BMD score, and TRAP5 in GD cohort. (**A**,**B**) Scatterplot analysis of age and Z- or T-score. (**C**,**D**) Scatterplot analysis of age and TRAP5b (**B**) and TRAP5a (**C**) showed no correlation between age and biomarkers. (**E**,**F**) The correlation between TRAP5b and Z-score (**E**) and TRAP5b and T-score (**F**). TRAP5b and Z-score were determined by linear Pearson correlation analysis. r = −0.4, *p* < 0.05. was considered statistically significant, with medium correlation. (**G**,**H**) A scatterplot analysis of TRAP5a and Z-score (**G**) or T-score (**H**) showed no correlation between biomarkers.

**Figure 5 cells-13-00716-f005:**
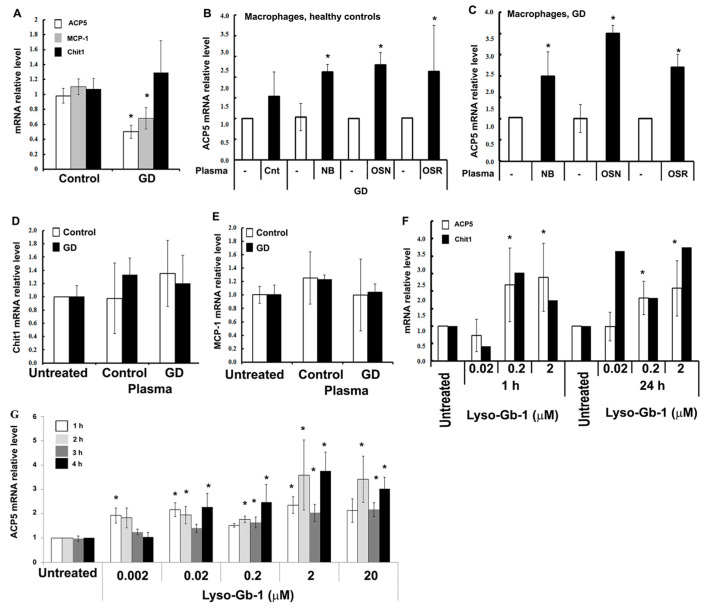
Expression of ACP5, Chit1, and MCP-1 mRNA in macrophages. (**A**) Basal levels of ACP5, MCP1, and Chit1 in control (n = 9) vs. GD macrophages (n = 9) were measured by q-RT-PCR. Values are averages +/− SEM. *t*-test *p* < 0.005. (**B**) Macrophages derived from healthy control PBMCs were treated with healthy control plasma (n = 5), and plasma collected from GD patients with normal bone mineral density (NB, n = 3), osteopenia (OSN, n = 4), and osteoporosis (OSR, n = 2). ACP5 was measured by q-RT-PCR. Values are averages +/− SEM. (**C**) Macrophages derived from GD PBMC were treated with plasma collected from GD patients with normal bone mineral density (NB), osteopenia (OSN), and osteoporosis (OSR). ACP5 was measured by q-RT-PCR. Values are averages +/− SEM. (**D**,**E**) Macrophages derived from healthy control and GD PBMCs were treated with plasma collected from healthy controls (n = 4) and GD patients (n = 3). q-RT-PCR measured Chit1 (**D**) and MCP-1 (**E**). Values are averages +/− SEM. (**F**) Macrophages derived from GD PBMCs were treated with increasing concentrations of Lyso-Gb-1 for 1 and 24 h. ACP5 was measured by q-RT-PCR. Values are averages +/− SEM. (**G**) THP-1 cells were treated with the indicated concentrations of Lyso-Gb-1, or vehicle control, for 1, 2, 3, and 4 h, and q-RT-PCR of ACP5 was performed. * Statistically significant differences.

**Table 1 cells-13-00716-t001:** The clinical features of GD patients, including bone diseases, plasma levels of TRAP5b and TRAP5a, inflammatory markers (monocyte chemoattractant protein 1 (MCP1), granulocyte macrophage colony-stimulating factor (GM-CFS), tumor necrosis factor-alpha (TNF-alpha), and GD biomarkers (CCL18, Lyso-Gb-1, Chito). NB—normal BMD, OSN—osteopenia, OSR—osteoporosis. Healthy control (n = 5). * Plasma was used for the treatment of macrophages in vitro.

	Controls	P1	P2 *	P15	P17	P23 *	P35 *	P37 *
GBA		N370S/ N370S	L444P/ L444P	N370S/ N370S	N370S/ N370S	N370S/ N370S	N370S/ N370S	N370S/ Y412X
Bone pathology	NB	NB	NB	OSN	OSN	OSN	OSR	OSR
TRAP5b	1.47 ± 0.09	2.0 ± 0.3	2.7 ± 0.1	1.7 ± 0.8	4.9 ± 2.6	2.4 ± 0.2	9.2 ± 4	7.7 ± 2
TRAP5a	5.1 ± 1.1	7.3 ± 1.3	6.5 ± 1.6	5.7 ± 1.9	62.8 ± 16	5.0 ± 1.6	7.7 ± 1.8	10.1 ± 1.1
CCL2/MCP1	27.18 ± 4.9	74 ± 6.2	45.7 ± 6.4	141.5	19 ± 7.2	189 ± 28	27 ± 17	68 ± 16
GM-CSF	1.9 ± 0.3	2.0 ± 0.3	1.7	0.2	1.4	76.5 ± 5	1.8	10.3
TNF-alpha	12.2 ± 2.9	33.8 ± 6.4	28.1	25 ± 3.7	32 ± 5.7	31 ± 5.7	23 ± 17	18 ± 3.9
CCL18	71 ± 20	60.9	180	37.3	407.0	97.5	331	562.9
Lyso-Gb1	normal	2.7	15.9	1.0	49.0	5.1	39	85.6
Chito	95 ± 26	52	273	42.0	1194.0	164.0	97.8	546.0

## Data Availability

Data are contained within the article and Appendix A.

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
