# Peer review of "The Expression and Secretion Profile of TRAP5 Isoforms in Gaucher Disease"

_cells, 2024, doi:10.3390/cells13080716_

Round 1
Reviewer 1 Report
Comments and Suggestions for Authors
1) No Correlation Between the Elevated Levels of TRAP5a and TRAP5b in GD
The above title is confusing, as further it is stated that “similar to our previous data [18], elevated levels of TRAP5b in GD correlate with osteopenia-osteoporosis”.
2) I recommend providing a graph including average values and SD for TRAP5a and TRAP5b levels in all groups below and check the significance while comparing to the control:
healthy controls,
GD patients treatment-naïve (N) if available,
GD patients treatment-naïve (OSN) if available,
GD patients treatment-naïve (OSR) if available,
GD patients on enzyme replacement therapy (N) if available,
GD patients on enzyme replacement therapy (OSN) if available,
GD patients on enzyme replacement therapy (OSR) if available,
GD on substrate reduction therapy (N) if available,
GD on substrate reduction therapy (OSN) if available,
GD on substrate reduction therapy (OSR) if available,
GD with a switch on ERT and SRT (N) if available,
GD with a switch on ERT and SRT (OSN) if available,
GD with a switch on ERT and SRT (OSR) if available.
3) TRAP5a and TRAP5b Monitoring Over 24 Months in Patients with GD
Is there a biological or a statistical difference in all groups mentioned above?
4) Please site the below papers, after this sentence:
Next, we compare TRAP5a and TRAP5b with Lyso-Gb1, the GD's most statistically 217 reliable diagnostic and pharmacodynamics biomarker [29].
https://pubmed.ncbi.nlm.nih.gov/29053611/
https://pubmed.ncbi.nlm.nih.gov/32605119/
5) “As an early indicator of BMD alteration, measurement of circulating TRAP5b is a valuable tool for assessing osteopenia-osteoporosis in GD”. Please add to the discussion the mRNA ACP5 levels and TRAP5b isoform levels in controls vs. Gaucher patents treatment naïve vs. Gaucher patients under treatment.
6) For the discussion: As Lyso-Gb1 directly increases ACP5 expression and Lyso-Gb1 levels correlate with TRAP5b levels, suggesting that the higher LysoGb1 the more sever the bone pathology is. Please add your explanation on what is the benefit to know TRAP5b levels in addition to LysoGb1 levels.
7) Standardize the results section titles. It is sometimes descriptive and sometimes gives a conclusion.
No Correlation Between the Elevated Levels of TRAP5a and TRAP5b in GD
TRAP5a and TRAP5b Monitoring Over 24 Months in Patients with GD
Correlation Analysis between TRAP5a, TRAP5b, and Inflammatory Biomarkers
TRAP5b, not TRAP5a, Correlates with the Z-score of Bone Mineral Density
ACP5 mRNA Expression in GD Macrophages in Vitro
Lyso-Gb-1 Increased ACP5 Expression in THP-1 Macrophages
Author Response
We would like to express our gratitude to the reviewer for carefully reading our manuscript and providing valuable feedback for its enhancement. We have addressed each of the comments and questions raised.
1) No Correlation Between the Elevated Levels of TRAP5a and TRAP5b in GD. The above title is confusing, as further it is stated that “similar to our previous data [18], elevated levels of TRAP5b in GD correlate with osteopenia-osteoporosis”.
Response: The title for the results section 3.1 has been changed.
2) I recommend providing a graph including average values and SD for TRAP5a and TRAP5b levels in all groups below and check the significance while comparing to the control:
healthy controls,
GD patients treatment-naïve (N) if available,
GD patients treatment-naïve (OSN) if available,
GD patients treatment-naïve (OSR) if available,
GD patients on enzyme replacement therapy (N) if available,
GD patients on enzyme replacement therapy (OSN) if available,
GD patients on enzyme replacement therapy (OSR) if available,
GD on substrate reduction therapy (N) if available,
GD on substrate reduction therapy (OSN) if available,
GD on substrate reduction therapy (OSR) if available,
GD with a switch on ERT and SRT (N) if available,
GD with a switch on ERT and SRT (OSN) if available,
GD with a switch on ERT and SRT (OSR) if available.
Response: We added correlation analysis between TRAP5a or TRAP5b and therapies in GD patients with normal BMD, osteopenia, and osteoporosis to Figure 1, including two more graphs (Figure 1 C and F). The results and Figure legend were updated: lines 168-191 (Results) and 196-216 (Figure legend).
3) TRAP5a and TRAP5b Monitoring Over 24 Months in Patients with GD.
Is there a biological or a statistical difference in all groups mentioned above?
Response: The following results show the trend of two biomarkers over a period of two years in individual patients. This data is unique to each patient and helps to determine if these biomarkers can be used as a dynamic marker for risk prediction. In most cases, data could divided into two groups based on the level of biomarkers: the level is normal for 2 years or elevated for 2 years. This pattern indicates that an elevation in biomarkers is more relevant to GD than other factors, such as a short transient immune response.
4) Please site the below papers, after this sentence: Next, we compare TRAP5a and TRAP5b with Lyso-Gb1, the GD's most statistically 217 reliable diagnostic and pharmacodynamics biomarker [29].
https://pubmed.ncbi.nlm.nih.gov/29053611/
https://pubmed.ncbi.nlm.nih.gov/32605119/
Response: Both manuscripts related to Lyso-Gb1 as a biomarker were included in the text.
5) “As an early indicator of BMD alteration, measurement of circulating TRAP5b is a valuable tool for assessing osteopenia-osteoporosis in GD”. Please add to the discussion the mRNA ACP5 levels and TRAP5b isoform levels in controls vs. Gaucher patents treatment naïve vs. Gaucher patients under treatment.
Response:
Due to the small size of our NAÏVE cohort, we did not discuss the relationship between the TRAP5b isoform and ACP5 mRNA level in NAÏVE versus treated patients. Only three patients in our study were NAÏVE, which is not sufficient for statistical analysis or drawing any conclusions. Therefore, we only provided an observational description.
We have included a discussion of the expression ACP5 in the 'Discussion' section.
6) For the discussion: As Lyso-Gb1 directly increases ACP5 expression and Lyso-Gb1 levels correlate with TRAP5b levels, suggesting that the higher LysoGb1 the more sever the bone pathology is. Please add your explanation on what is the benefit to know TRAP5b levels in addition to LysoGb1 levels.
Response: Discussion of benefits of knowledge of TRAP5b level in addition to Lyso-Gb-1 in GD were included in the text.
7) Standardize the results section titles. It is sometimes descriptive and sometimes gives a conclusion.
No Correlation Between the Elevated Levels of TRAP5a and TRAP5b in GD TRAP5a and TRAP5b Monitoring Over 24 Months in Patients with GD
Correlation Analysis between TRAP5a, TRAP5b, and Inflammatory Biomarkers TRAP5b, not TRAP5a,
Correlates with the Z-score of Bone Mineral Density ACP5 mRNA Expression in GD Macrophages in Vitro Lyso-Gb-1 Increased ACP5 Expression in THP-1 Macrophages
Response: The results were standardized.
Reviewer 2 Report
Comments and Suggestions for Authors
The manuscript by Ivanova and colleagues describes a detailed analysis of the TRAP5 isoforms in a cohort of selected Gaucher disease patients affected by a variable degree of bone abnormalities, ranging from normal BMD to ostenecrosis and osteoporosis. The Authors next provided preliminary data regarding the evaluation of a potential correlation between TRAP5 isoforms levels with those of known GD biomarkers, including CCL18, Lyso-Gb and chitotriosidase.
While the manuscript provides quite interesting data I personally found some relevant issues that need to be addressed before considering the paper acceptable for publication.
My first concern regards the language and the general writing. I found many relevant mistakes in several sentences, including conceptual errors in the abstract (isotopes and isomers instead of isoforms).
- In the abstract beyond the wrong terms isomers and isotopes please add plasma levels in lane 25
- Lane 38: add “to the” before deficiency or better adjust the sentence
- Lane 45-46: please use a more correct term instead of degrade. Gb-1 is degraded or catabolyzed. It doesn’t degrade.
- Lane 58-60: adjust the sentence by adding “consisting of a” to postranslational…
- Lane 66: an ELISA-based immunodetection assay instead of ELISA kit
- Lane73-75: please rewrite the sentence
- Lane 84: please specify the number of purchased plasma samples of healthy controls
- Lane 90-94 : adjust and rewrite the sentences
- Lane 103 “PBMCs were” instead of PBMC was
- Lane 122: “samples” instead of “cells”
- Lane 127: GAPDH is not a loading control; but iot is rather the housekeeping gene
- Lane 218-221: adjust and rewrite the sentence
- Lane 349-350: adjust and rewrite the sentence
- Lane 378-379: adjust and rewrite the sentence
- In Fig.1A an F-test was used while in Fig 1C a t-test was used. I assume Authors have chosen the F-test because the t-test gave a non-statistically significant result. I would strongly recommend to remove the F-test and just simply say there is no statistically significant increase in TRAP5a levels in GD patients. This would match with further results of cross-correlation with bone abormalities for only TRAP5b. Then, all related comments on Result and Discussion sections should be adjusted, including a general "tune-down" of the role of TRAP5a as a relevant biomarker for GD, given the lack of its significant changes in tested samples.
- Fig1D: correct INOVA.
- In FIg 5D there is a mismatch with the legend. Is that ACP5 or Chit?
- In Fig 5G the annotation is missing. And please add statistics.
- In all Figures Legends with RQ-PCRS there is lack of technical replicates number.
Authors should also further include in the Discussion section a comparison of the their data with those of Deegan and colleagues (2010, Blood Cells, Molecules, and Diseases who mentioned TRAP activity cannot be considered as a prognostic biomarker.
Comments on the Quality of English LanguageAs written in the Suggestions, I would encourage Authors to thoroughly revise the text and address all relevant concerns.
Author Response
We would like to express our gratitude to the reviewer for carefully reading our manuscript and providing valuable feedback for its enhancement. We have addressed each of the comments and questions raised.
Comments and Suggestions for Authors
The manuscript by Ivanova and colleagues describes a detailed analysis of the TRAP5 isoforms in a cohort of selected Gaucher disease patients affected by a variable degree of bone abnormalities, ranging from normal BMD to osteonecrosis and osteoporosis. The Authors next provided preliminary data regarding the evaluation of a potential correlation between TRAP5 isoform levels with those of known GD biomarkers, including CCL18, Lyso-Gb, and chitotriosidase.
While the manuscript provides quite interesting data I personally found some relevant issues that need to be addressed before considering the paper acceptable for publication.
My first concern regards the language and the general writing. I found many relevant mistakes in several sentences, including conceptual errors in the abstract (isotopes and isomers instead of isoforms).
Response: We replace isotopes and isomers to isoforms.
1 In the abstract beyond the wrong terms isomers and isotopes please add plasma levels in lane 25
Response: “Plasma” word was added to the abstract.
2 Lane 38: add “to the” before deficiency or better adjust the sentence
Response: The sentence was corrected.
3 Lane 45-46: please use a more correct term instead of degrade. Gb-1 is degraded or catabolized. It doesn’t degrade.
Sentence: Due to a GCase enzyme deficiency, the last glycolipid in the catabolic pathway of the glycosphingolipid metabolism, glucosylceramide (Gb-1), does not completely degrade in GD, have been changed.
4 Lane 58-60: adjust the sentence by adding “consisting of a” to posttranslational…
5 Lane 66: an ELISA-based immunodetection assay instead of ELISA kit
Response: The sentence was changed.
6 Lane73-75: please rewrite the sentence
Response: The sentence was rewritten.
7 Lane 84: please specify the number of purchased plasma samples of healthy controls
Response: The number of control samples has been added to the legend of Figure 1.
8 Lane 90-94 : adjust and rewrite the sentences
Response: We made adjustments to the sentences.
9 Lane 103 “PBMCs were” instead of PBMC was
Response: The sentence was changed.
10 Lane 122: “samples” instead of “cells”
Response: If we understand correctly, that issue is with line 122. Line 122 describes experiments with macrophages and the THP cell line. Therefore, using the word “cells” is correct.
11 Lane 127: GAPDH is not a loading control; but iot is rather the housekeeping gene
Response: The control used in experiments has been switched from "loading control" to "reference genes. “Housekeeping genes is another term that refers to the same thing. In my opinion, "housekeeping genes" is an odd term because it implies cleaning a house in general. The term "loading control" is an “old school” term that was used before real-time PCR.
12 Lane 218-221: adjust and rewrite the sentence
Response: New sentence: “There was a strong positive correlation (r=0.8) between TRAP5b and Lyso-Gb1, while no correlation was found between TRAP5a and Lyso-Gb1 (r=0.002) (Figure 3 C and D-F).”
13 Lane 349-350: adjust and rewrite the sentence
Response: The sentence was adjusted.
14 Lane 378-379: adjust and rewrite the sentence
Response: We added a new sentence: “ Therefore, TRAP5a/Chito can be viable biomarkers to verify macrophage activation in the presence of immune inflammation in GD”
15 In Fig.1A an F-test was used while in Fig 1C a t-test was used. I assume Authors have chosen the F-test because the t-test gave a non-statistically significant result. I would strongly recommend to remove the F-test and just simply say there is no statistically significant increase in TRAP5a levels in GD patients. This would match with further results of cross-correlation with bone abormalities for only TRAP5b. Then, all related comments on Result and Discussion sections should be adjusted, including a general "tune-down" of the role of TRAP5a as a relevant biomarker for GD, given the lack of its significant changes in tested samples.
Response: The statistical analysis details, including unpaired T-test and F-test, are described in the materials and methods and Figure legends. Graph Prizm analysis coupled these tests and confirmed that the statistical differences between TRAP5a levels in Control and GD are significant (as shown in the table below). Additionally, Fig 1A legends described both tests:
“TRAP5a level, Control vs. GD. * F-Test, two tail P<0.0001, Unpaired T-test one tail P= 0.0323 P<0.05, two tail P=0.0646, Control n= 35, GD n=39.’.
16 Fig1D: correct INOVA.
17 In FIg 5D there is a mismatch with the legend. Is that ACP5 or Chit?
Response: The typos were fixed.
18 In Fig 5G the annotation is missing. And please add statistics.
Response: The Fig 5 was updated.
19 In all Figures Legends with RQ-PCRS there is lack of technical replicates number.
Response: We have added information to the “Material and Methods” section. To determine gene expression, all experiments were carried out in triplicates, and the results were using GAPDH.
Moreover, in Figure 5, we have clarified the sample size used in the experiment. We differentiated macrophages from 5 healthy controls and 9 patients with GD.
20 Authors should also further include in the Discussion section a comparison of the their data with those of Deegan and colleagues (2010, Blood Cells, Molecules, and Diseases who mentioned TRAP activity cannot be considered as a prognostic biomarker.
Response: Unfortunately, I was not able to find Patrick Deegan's 2010 publication related to the TRAP biomarker in GD. Instead, only a 2011 publication about osteonecrosis and inflammatory biomarkers in GD was found. However, this MS is focusing on osteonecrosis and not on osteoporosis and bone mineral density.
“Potential biomarkers of osteonecrosis in Gaucher disease” Elena V Pavlova 1, Patrick B Deegan, Jane Tindall, Ian McFarlane, Atul Mehta, Derralyn Hughes, J Edmond Wraith, Timothy M Cox’’.
Reviewer 3 Report
Comments and Suggestions for Authors
Manuscript ID: cells-2903949
Article ‘The Expression and Secretion Profile of TRAP5 Isoforms in Gaucher Disease’
Authors: Margarita M Ivanova et al.
This is a very interesting manuscript on the different ways of activation of TRAP5 isoforms in patients with Gaucher disease. Authors have examined expression of TRAP5a and TRAP5b in the material from Gaucher patients at different age, with or without therapy and control individuals.TRAP5a is an inflammatory biomarker, while TRAP5b is a bone biomarker secreted from osteoclasts.
Obtained results indicate that levels of TRAP5a and TRAP5B did not depend on the age of patients. An increased level of TRAP5a in GD is not correlated with TRAP5b or decreasing bone mineral density (BMD) and thus does not correlate with osteoclast activity and acceleration of bone resorption. The up-regulated expression of TRAP5b was associated with decreasing BMD and thus it can potentially serve as a valuable tool for assessing osteopenia-osteoporosis in GD. The up-regulation of TRAP5a is rather an indicator of macrophages' pro-inflammatory activities.
Although the manuscript is written interestingly, it will improve after English editing. I have also found some other points:
Abstract
- ‘ACP5’ should be explained
- P.1, line 28-29 ‘The expression of ACP5 in macrophages is regulated by plasma and Lyso-Gb1.’ is not clear.
Introduction
- Usually, three forms of GD are distinguished not two: non-neuronopathic, severe neuronopathic and subacute neuronopathic.
Materials and Methods
- ‘IRB-approved’ should be explained
- The nosology of genetic variants (mutations) is traditional. It should be indicated in the text that for practical reasons mutation e.g. p.N370S will be named N370S further in the text.
- P.2, line 93 – BMB should be BMD
- PART ‘2.2. Blood Sample Collection’ is written very generally. Please, indicate the volume of blood sample, conditions of centrifugation, what served for isolation of PBMC (plasma? The residual material with morphotic elements?), were isolated PBMC also frozen or used immediately for experiments?
- Part ‘2.3. Differentiation of M2-Macrophages from PBMC’ – please, indicate conditions of incubation – temperature CO2 percentage, etc.
- Part 2.4. - The abbreviation ‘THP-1 cell line’ should be explained and also why these cells were used in experiments, why PMA was added to the culture medium (what is the action of this compound); in this paragraph also temperature of incubation and CO2 percentage in the air are needed to be added
- Line 124 ‘Quantitative Real-Time-PCR (qPCR)’ I suggest to use qRT-PCR abbreviation
- Part 2.6. ‘GADPH’ should be GAPDH
- Part 2.8. Please rewrite the sentence ‘The concentrations of the substrates 4-methylumbelliferyl-β-D-146 N,N′,N″-triacetyl-chitotrioside (4MU-C3, Sigma®) in 0.1 M/0.2 M citrate-phosphate 147 buffer.’. In the present form it is not clear.
Results
- Part ‘3.2. TRAP5a and TRAP5b Monitoring Over 24 Months in Patients with GD’ - is it possible to see any differences in TRAP5a and TRAP5b levels depending on the kind of therapy used in GD patients?
- Fig.3F – Spearman
- In part 3.5. and in the whole text the nosology of genes is mixed e.g. CHIT1, Chit1, CHIT1 . Please, standardize the notation.
Legend to Fig.1, line 180, should be NB.
On Fig.1.D. should be ANOVA.
On Fig.3F. should be Spearman.
In the legend to Table 1 the abbreviations MCP1, GM-CFS, TNF-alpha should be explained.
In the legend to Fig.5 – should be qRT-PCR. What is ‘average +/- STEV’ ?
Comments on the Quality of English Language
Minor editing of English will improve the manuscript.
Author Response
We would like to express our gratitude to the reviewer for carefully reading our manuscript and providing valuable feedback for its enhancement. We have addressed each of the comments and questions raised.
This is a very interesting manuscript on the different ways of activation of TRAP5 isoforms in patients with Gaucher disease. Authors have examined expression of TRAP5a and TRAP5b in the material from Gaucher patients at different age, with or without therapy and control individuals. TRAP5a is an inflammatory biomarker, while TRAP5b is a bone biomarker secreted from osteoclasts.
Obtained results indicate that levels of TRAP5a and TRAP5B did not depend on the age of patients. An increased level of TRAP5a in GD is not correlated with TRAP5b or decreasing bone mineral density (BMD) and thus does not correlate with osteoclast activity and acceleration of bone resorption. The up-regulated expression of TRAP5b was associated with decreasing BMD and thus it can potentially serve as a valuable tool for assessing osteopenia-osteoporosis in GD. The up-regulation of TRAP5a is rather an indicator of macrophages' pro-inflammatory activities. Although the manuscript is written interestingly, it will improve after English editing. I have also found some other points:
Abstract
- ‘ACP5’ should be explained
Response: Explanation that ACP5 is a gene abbreviation and TRAP is a protein abbreviation included to the abstract.
- P.1, line 28-29 ‘The expression of ACP5 in macrophages is regulated by plasma and Lyso-Gb1.’ is not clear.
Response: New sentence: “Lyso-GB-1 and plasma have the ability to control the expression of ACP5 in macrophages:
Introduction
- Usually, three forms of GD are distinguished not two: non-neuronopathic, severe neuronopathic and subacute neuronopathic.
Response: We changed the sentence.
Materials and Methods
- ‘IRB-approved’ should be explained
Response: We added IRB abbreviation.
- The nosology of genetic variants (mutations) is traditional. It should be indicated in the text that for practical reasons mutation e.g. p.N370S will be named N370S further in the text.
Response: We added p.N370S and p.L444P in material and methods.
- P.2, line 93 – BMB should be BMD
Response: The typo was fixed.
- PART ‘2.2. Blood Sample Collection’ is written very generally. Please, indicate the volume of blood sample, conditions of centrifugation, what served for isolation of PBMC (plasma? The residual material with morphotic elements?), were isolated PBMC also frozen or used immediately for experiments?
Response: We made updates to the "Sample collection" text. Some PBMCs were used to differentiate macrophages, while others were frozen.
- Part ‘2.3. Differentiation of M2-Macrophages from PBMC’ – please, indicate conditions of incubation – temperature CO2 percentage, etc.
Response: We added to the text: All the cell culture experiments be maintained at 37°C and 5% CO2.
- Part 2.4. - The abbreviation ‘THP-1 cell line’ should be explained and also why these cells were used in experiments, why PMA was added to the culture medium (what is the action of this compound); in this paragraph also temperature of incubation and CO2 percentage in the air are needed to be added
Response: We updated the material and methods: The THP1 cell line is a human monocytic cell line derived from an acute monocytic leukemia patient. It can be differentiated into macrophages in the presence of PMA.
- Line 124 ‘Quantitative Real-Time-PCR (qPCR)’ I suggest to use qRT-PCR abbreviation
Response: The abbreviation has been changed.
- Part 2.6. ‘GADPH’ should be GAPDH
Response: The typo has been fixed.
- Part 2.8. Please rewrite the sentence ‘The concentrations of the substrates 4-methylumbelliferyl-β-D-146 N,N′,N″-triacetyl-chitotrioside (4MU-C3, Sigma®) in 0.1 M/0.2 M citrate-phosphate 147 buffer.’. In the present form it is not clear.
Response: the sentence was clarify.
Results
- Part ‘3.2. TRAP5a and TRAP5b Monitoring Over 24 Months in Patients with GD’ - is it possible to see any differences in TRAP5a and TRAP5b levels depending on the kind of therapy used in GD patients?
Response: We believe that both biomarkers are useful for monitoring patients over time and evaluating the effectiveness of therapies.
- Fig.3F – Spearman
Response: the typo was fixed.
- In part 3.5. and in the whole text the nosology of genes is mixed e.g. CHIT1, Chit1, CHIT1 . Please, standardize the notation.
Response: we replace CHIT1 to Chit1.
- Legend to Fig.1, line 180, should be NB.
- On Fig.1.D. should be ANOVA.
- On Fig.3F. should be Spearman.
Response: We fixed the typo.
- In the legend to Table 1 the abbreviations MCP1, GM-CFS, TNF-alpha should be explained.
Response: Protein names we included.
In the legend to Fig.5 – should be qRT-PCR. What is ‘average +/- STEV’ ?
Response: the sentences were fixed.
Round 2
Reviewer 2 Report
Comments and Suggestions for Authors
The revised version of the manuscript by Ivanova and colleagues has addressed most of my previously raised concerns. While in general I appreciate the significant improvement of the paper there are still required changes.
- Line 80-81: there are two typos.The first relates to the term “its”. I suppose it should changed into “them”.The second one is “funding” instead of finding
- Mat & Methods: Line 99-100 Patients into the three categories are 40 (12+14+14) and not 39. If there is a patient that is shared between two categories this needs to be clarified.
- Line 106: spin needs to be changed into “spun”
- Figure 1E and 1F and Figure 1 Legend: There is still “INOVA” related to Kruskal-Wallis test. To my knowledge the KW test is a One-Way ANOVA.
- Line 228: Pearson instead of Person.
- Legend of Fig 4. I do apologize but I missed this point in the previous version. Why Authors are stating that “p<0.1 in the correlation was considered statistically significant”. Generally the values of r :–0.30 indicate a weak negative linear relationship. Can Authors explain better this point?
- Line 325: decreased instead of decrease
- Finally once again my apologies, but the paper of Deegan was referring to “Deegan PB, Moran MT, McFarlane I, Schofield JP, Boot RG, Aerts JM, Cox TM. Clinical evaluation of chemokine and enzymatic biomarkers of Gaucher disease”, where in the Introduction was written “TRAP is neither specific for Gaucher disease nor greatly elevated”. While I suppose that this old concept was due to the lack of sensitivity of previous assays, addressing a final comment in the Discussion or conclusion would help readers to clarify the paradigm change in the context of TRAP as a new valuable clinical biomarker in GD.
Comments on the Quality of English Language
Already reported in my comments to Authors. Minor spell check required.
Author Response
The second round of revision.
Reviewer 2.
We would like to express our gratitude to Reviewer 2 for their thorough review of our manuscript and for providing valuable feedback. We have responded to each of the comments/questions in your letter and in those from each reviewer in Arial font 11 (not bold) size below with the original letter and reviews in Times-New Roman (bold, blue).
We hope that the revised manuscript will be acceptable for publication.
Comments and Suggestions for Authors
The revised version of the manuscript by Ivanova and colleagues has addressed most of my previously raised concerns. While in general I appreciate the significant improvement of the paper there are still required changes.
- Line 80-81: there are two typos.The first relates to the term “its”. I suppose it should changed into “them”.The second one is “funding” instead of finding.
Response: Thank you very much for catching the typos; typos have been fixed.
- Mat & Methods: Line 99-100 Patients into the three categories are 40 (12+14+14) and not 39. If there is a patient that is shared between two categories this needs to be clarified.
Response: One patient was excluded from the study. We fixed the number of patients.
- Line 106: spin needs to be changed into “spun”
Response: The sentence: “The plasma was then aliquoted into 1.5 mL tubes and spin down at 2000xg for 2 minutes to clear samples”. We changed the spin to spun.
- Figure 1E and 1F and Figure 1 Legend: There is still “INOVA” related to Kruskal-Wallis test. To my knowledge the KW test is a One-Way ANOVA.
- Line 228: Pearson instead of Person.
Response: We fixed mistakes and uploaded a new Figure 1.
- Legend of Fig 4. I do apologize but I missed this point in the previous version. Why Authors are stating that “p<0.1 in the correlation was considered statistically significant”. Generally the values of r :–0.30 indicate a weak negative linear relationship. Can Authors explain better this point?
Response: Thank you for pointing out the mistake. After several revisions to the figures and figure legends with track changes, the text became unclear and lost its meaning. We reanalyzed the data and updated Figure 4 and Figure Legend. Below, the table represents Graph Pad Prism statistical analysis related to the correlation between the Z score and TRAP5b level in patients with GD. Z-scores (26 patients) and T-scores (13 patients) were used based on patient age for a total of 39 patients.
We acknowledge that a correlation coefficient of -0.4 is not a perfect correlation, but with a small number of pairs for correlation analysis, it is still considered a "medium correlation". While a correlation value of 0 indicates no correlation.
https://www.statisticssolutions.com/free-resources/directory-of-statistical-analyses/pearsons-correlation-coefficient/
- Line 325: decreased instead of decrease
Response: We fixed mistakes
- Finally once again my apologies, but the paper of Deegan was referring to “Deegan PB, Moran MT, McFarlane I, Schofield JP, Boot RG, Aerts JM, Cox TM. Clinical evaluation of chemokine and enzymatic biomarkers of Gaucher disease”, where in the Introduction was written “TRAP is neither specific for Gaucher disease nor greatly elevated”. While I suppose that this old concept was due to the lack of sensitivity of previous assays, addressing a final comment in the Discussion or conclusion would help readers to clarify the paradigm change in the context of TRAP as a new valuable clinical biomarker in GD.
Response: The reference “Deegan PB, et al. Clinical evaluation of chemokine and enzymatic biomarkers of Gaucher disease” has been added to the text.